



# Deep Learning Emulation of Multivariate Climate Indices: A Case Study of the Fire Weather Index in the Iberian Peninsula

Óscar Mirones[1], Joaquín Bedia[2,3], Pedro M.M. Soares[4], José M. Gutiérrez[1], and Jorge Baño-Medina[1,5]

[1]Instituto de Física de Cantabria (IFCA), CSIC-Universidad de Cantabria, Santander, Spain

[2]Dept. Matemática Aplicada y Ciencias de la Computación (MACC), Universidad de Cantabria, Santander, Spain

[3]Grupo de Meteorología y Computación, Universidad de Cantabria, Unidad Asociada al CSIC, Santander, Spain

[4]Instituto Dom Luiz (IDL) - Faculdade de Ciências da Universidade de Lisboa (FCUL), Campo Grande Edifício C8, Piso 3, 1749-016 Lisboa

[5]Center for Western Weather and Water Extremes, Scripps Institution of Oceanography, University of California San Diego, San Diego, CA, USA

**Correspondence:** Jorge Baño-Medina (bmedina@ifca.unican.es)

**Abstract.** The Fire Weather Index (FWI) is an essential multivariate climate index for assessing wildfire risk and the associated impacts of climate change, as it provides a quantitative measure of wildfire danger by integrating different critical near-surface fire-weather variables, namely air temperature, relative humidity, wind speed, and precipitation. FWI calculation depends on instantaneous data representing noon local standard times, which are often unavailable in many climate data repos-
itories—particularly in climate projections. In these instances, a "proxy" of actual FWI is often used, applying the same FWI formulation to daily aggregated values (mean, max, or min), despite known limitations in capturing extremes and temporal dynamics.

This study investigates the use of deep learning (DL) models to emulate the reference FWI over the Iberian Peninsula—a predominantly Mediterranean and fire-prone region—using only daily inputs. The emulators are trained and evaluated using
ERA5-Land data, which, while not observational ground truth, provides a consistent and high-resolution dataset suitable for controlled inter-comparison. The focus is not on validating FWI against observations, but on assessing the ability of DL models to reproduce the reference FWI more accurately than traditional proxy approaches, using the same input data source.

Our results show substantial improvements in spatial accuracy, preservation of temporal sequences, and detection of extreme fire danger events when compared with the corresponding proxy version. Furthermore, after evaluating different com-
binations of input variables for DL model training, we find that precipitation can be excluded without substantially affecting accuracy—especially at the upper end—an important insight given the challenges climate models face in representing precipitation. These findings highlight the potential of deep learning tools to enhance the usability of FWI in contexts where sub-daily data are unavailable, and set the stage for the emulation of other multivariate climate indices, which are vital for climate impact studies, spatial planning and management, and adaptation decision-making.



## 1 Introduction

Fire is a global phenomenon that has shaped ecosystems since the emergence of land vegetation, influencing vegetation structure, the carbon cycle, and climate (Bowman et al., 2009). From a human perspective, wildfires can cause severe ecological damage, threaten lives and infrastructure, and degrade air and water quality, with far-reaching social and economic consequences (Bowman et al., 2017; Pausas and Keeley, 2021).

Fire danger indices are essential tools for wildfire risk assessment and climate impact studies. By integrating key atmospheric variables—such as temperature, humidity, wind speed, and precipitation—into a single, physically interpretable indicator (Fugioka et al., 2009; Yu et al., 2023), these indices support fire management agencies in resource allocation and early warning systems (Di Giuseppe et al., 2020). They also help researchers understand how climate change alters fire-weather patterns (Bedia et al., 2015; Jolly et al., 2015; Bento et al., 2023; Santos et al., 2023; Gincheva et al., 2024) informing adaptation strategies and policy decisions in fire-prone regions (Costa et al., 2020; DaCamara et al., 2024).

Among the various fire-weather indices worldwide, the Canadian Fire Weather Index (FWI, van Wagner, 1987) is one of the most widely adopted globally. In Europe, it underpins the European Forest Fire Information System (EFFIS San-Miguel-Ayanz et al., 2013) and has been extensively used for the estimation of fire danger in vulnerable regions (Camia and Amatulli, 2009), such as Iberia (Santos et al., 2023), as well as for the assessment of future climate change scenarios in the Euro-Mediterranean region (Bedia et al., 2014; Dupuy et al., 2020) and the Iberian Peninsula (Bento et al., 2023).

Although the adoption of the FWI presents clear advantages for effectively characterizing fire danger weather situations, it also poses some hurdles from an implementation point of view. One significant challenge is that the FWI was originally designed for noon local standard time (LST) conditions, representing the highest fire danger time in mid-latitude regions. This means it requires instantaneous noon time input data of near-surface air temperature, relative humidity, 10-m wind speed and last 24-hour accumulated precipitation (Lawson and Armitage, 2008). However, such specific noon data are often missing from observational records. Furthermore, climate model databases quite often do not include sub-daily outputs due to the massive storage requirements (e.g. the Earth System Grid Federation –ESGF–, Cinquini et al., 2014), thus preventing the direct calculation of FWI from climate model simulations.

To circumvent the lack of instantaneous data, daily mean aggregated inputs have been commonly used for FWI estimation, in particular to obtain future projections from model simulations. However, the resulting daily mean FWI proxies cannot be reliably transformed to match their instantaneous counterparts, leading to inconsistent future projections (Herrera et al., 2013). Consequently, approximations are often employed to minimize FWI distortion. One of the pioneering studies in this area utilized different daily *proxy* variables for FWI calculation from climate model outputs, comparing various combinations of minimum relative humidity and maximum temperature as surrogates for noon-time outputs. This approach aimed to minimize the distortion of the climate change signal in the reference FWI, using simulations from a regional model for both historical and a moderate future warming scenario (Bedia et al., 2014). Since then, other studies have adopted proxies to produce more accurate FWI projections (Abatzoglou et al., 2018; Bento et al., 2023; Matteo et al., 2025), as a practical solution given the



unavailability of more precise data. Despite all these efforts, the approximate version of the FWI inevitably presents certain deviations from the original version that introduce added uncertainty (Bedia et al., 2014; Matteo et al., 2025).

Emulation offers a promising alternative in this context. Recent advancements in machine learning techniques, particularly deep learning (DL), across various fields suggest that these sophisticated nonlinear models have the potential to effectively represent and capture the intricate structures and nonlinear dynamics inherent in Earth systems (Duffy et al., 2023), including numerical model climate predictions (Rampal et al., 2024) and applications to downscaling (Baño-Medina et al., 2022; Bushenkova et al., 2024; Li et al., 2025; Johannsen et al., 2024; Soares et al., 2024), among others. In this new framework,
the emulation function is determined by the neural network's coefficients, allowing for varying topologies to produce different plausible fields (Baño-Medina et al., 2024). This opens up a broad area of research where various techniques and architectures can be compared to evaluate their suitability for specific problems. It also introduces new challenges related to model interpretability, requiring methods of eXplainable Artificial Intelligence (XAI) that reveal how DL models make decisions. Such approaches play a key role in establishing the credibility of the outcomes among users (Dramsch et al., 2025) and for
gaining valuable insights into the most influential variables and the physical interpretation of model's functioning (Yang et al., 2024; Baño-Medina et al., 2025). Among the variety of approaches available, saliency maps emerge as valuable tools aiding in the design and evaluation of DL climate downscaling approaches in general (González-Abad et al., 2023), and have been successfully used to unravel DL-based FWI predictions in previous studies (Mirones et al., 2025).

In this study, we explore the emulation of the Fire Weather Index (FWI) using DL models to approximate the index's
behavior as closely as possible with commonly available input variables whose temporal frequency (daily) does not match the definition of the "reference FWI" (based on instantaneous noon local standard times). We build upon the reference FWI and the optimal proxy FWI version presented in Bedia et al. (2014), which has been widely used in subsequent studies. We compare the emulation of the reference FWI using different combinations of input variables (hereafter predictor sets), including inadequate (daily mean), incomplete (removing some input variables), or suboptimal temporal frequency inputs (i.e.
daily minimum relative humidity), which are more commonly available than the required instantaneous data. We test various DL model architectures and topologies and utilize explainable AI (XAI) techniques, such as saliency maps, to provide end users and practitioners with a solid understanding of the emulation process. Our results indicate that DL emulators outperform the traditional proxy approach in all validation aspects, including spatial representation, preservation of the temporal sequence, and detection of extreme fire danger conditions. The daily mean inputs, which are most widely available in public repositories, are
adequate for accurately emulating the reference FWI using the developed DL methods. Furthermore, our findings suggest that precipitation can be omitted from the predictor set without significantly compromising FWI representation accuracy. Overall, we demonstrate that by leveraging simpler inputs, DL emulators can enhance the accessibility and applicability of FWI in impact studies, facilitating a more efficient and widespread use in climate impact studies and decision-making.



## 2 Data and Methods

### 2.1 Input Data

All data used in this study were obtained from the ERA5-Land database (Muñoz-Sabater et al., 2021), distributed by the Climate Data Store (CDS) of the Copernicus Climate Change Service (Buontempo et al., 2022). ERA5-Land is a high-resolution reanalysis dataset produced by the European Centre for Medium-Range Weather Forecasts (ECMWF). It offers detailed information on land surface variables, providing a consistent view of their evolution over several decades. ERA5-Land features a finer spatial resolution of approximately 9 km grid spacing, compared to ERA5 (∼25 km Hersbach et al., 2020). It covers data from January 1950 to the present, with an hourly temporal resolution. The dataset includes various land surface parameters (such as soil moisture, soil temperature, snow cover, and surface runoff) to control the simulated land fields, ensuring the data remains accurate and consistent. ERA5-Land uses atmospheric variables from ERA5, like air temperature, wind speed and humidity, and presents hourly temporal resolution, thus providing the necessary input variables for the calculation of both reference and proxy FWI versions, as well as the different predictor sets used in this study (Sec. 2.3).

### 2.2 Fire Weather Index calculation

The reference noon LST FWI is calculated from ERA5-Land daily records of four near-surface meteorological variables measured at 12 UTC (∼ noon time in the Iberian Peninsula): 24-hour accumulated precipitation, instantaneous wind speed, relative humidity, and temperature. These inputs are processed via a set of empirical equations that yield six intermediate components characterizing fuel moisture dynamics across different fuel layers (van Wagner, 1987; Stocks et al., 1989), the influence of wind on fire spread, and the total fuel available for combustion. These components are then integrated to compute the FWI, a dimensionless index representing potential fire intensity under given meteorological conditions for a reference fuel type (mature pine stands).

A key complexity arises from the fact that certain FWI components, particularly those related to fuel moisture, exhibit different drying rates and retain memory of past conditions (Wotton, 2009). To address this, FWI initialization relies on default values for some components, leading to a spin-up period before stabilization. This period is generally brief, typically lasting a few days to weeks, especially during the fire season, when the effects of snow melting on soil moisture are minimal. Consequently, spin-up effects are deemed negligible due to the rapid stabilization of the FWI over time and any residual impact of spin-up is expected to be minimal in the final outputs (Bedia et al., 2018).

### 2.3 Predictors Sets

To emulate the reference FWI, we consider several predictor sets derived from ERA5-Land, summarized in Table 1. The initial experiments use P0, which corresponds to the "Best" Proxy FWI previously evaluated against similar approaches in Bedia et al. (2014). The objective is to assess whether the DL models can more accurately replicate the FWI transfer function using the same predictors as the Proxy FWI, which is derived from standard FWI formulation. Since precipitation is one of



the most challenging variables to represent dynamically by climate models (including reanalysis) and to emulate —due to its high spatial and temporal variability, complex physical processes, and dependence on multiple interacting factors— we also evaluate predictor sets that exclude precipitation. Additionally, all predictor sets in this study consist solely of daily mean values (excluding 24h-accumulated precipitation in P0), avoiding sub-daily or instantaneous timescales, which are generally less often available than daily mean data. P1 and P2 are evaluated as alternative emulation inputs (Table 1), aiming to improve upon P0 results.

| Predictors sets | | Variable | | | | | |
|---|---|---|---|---|---|---|---|
| | Description | Temp. | Rel. Hum. | Precipitation | Wind speed | Eastward wind | Northward wind |
| FWI | Reference FWI definition (van Wagner, 1987) | 12UTC | 12UTC | 24-h accumulated (12UTC-12UTC) | 12UTC | × | × |
| P0 | "Best" Proxy FWI (Bedia et al., 2014) | DM | Min | 24-h accumulated (12UTC-12UTC) | DM | × | × |
| P1 | Daily Mean inputs without precipitation | DM | DM | × | DM | × | × |
| P2 | Daily Mean inputs with wind components | DM | DM | × | DM | DM | DM |

**Table 1.** Summary of the predictor sets assessed in this study. 12 UTC correspond to instantaneous model outputs verifying at that time. DM corresponds to Daily Mean values, and Max/Min to Daily Maximum/Minimum values. Precipitation is the daily (last 24 hours) accumulated value, from 12 to 12 UTC. Cells marked with × indicate that the variable is not included in the corresponding predictor set.

## 2.4 Deep Learning Methods

In this section, we briefly describe the DL architectures trained in this study. A visual summary of their topologies is presented in Fig. 1. The selected architectures include the Fully Connected Dense (FCD) model, chosen as benchmark for its relative simplicity, as well as DeepESD and U-Net, which have been recently introduced in the literature for climate downscaling (Baño-Medina et al., 2022; González-Abad et al., 2023; González-Abad and Gutiérrez, 2024). The DL models are tasked to learn a mapping between different sets of FWI-related input variables (Table 1) and the reference FWI over the Iberian Peninsula as output, by minimizing a loss function (Sec. 2.4.4). The model output is an emulated version of the reference ERA5-Land FWI at the same spatial resolution than its inputs. All models are trained with the same optimization parameters independently of the architecture, using the Adam optimizer with a learning rate set to 0.0001, a batch size of 64 and a maximum of 1,000 epochs. An early stopping mechanism is set to prevent overfitting, which stops the training process if there is no improvement in the validation loss after 30 consecutive epochs. Furthermore, the input data are standardized as part of the model preprocessing.





### 2.4.1 Fully Connected Dense

The Fully Connected Dense (FCD) model is a DL architecture that relies exclusively on fully connected layers to process spatial
input data. The input is first passed through two consecutive dense layers, each containing 50 neurons, with Rectified Linear
Unit (ReLU) activation functions (Glorot and Bengio, 2010) applied to introduce non-linearity and enhance feature extraction.
Following these hidden layers, the transformed feature representation is fed into a fully connected layer with 12,800 neurons,
utilizing a linear activation function to produce the final output. This output is then reshaped to match the dimensions of the
predictand, ensuring consistency with the target variable.

### 2.4.2 DeepESD

The DeepESD model (Baño-Medina et al., 2022) is designed as a combination of convolutional and dense layers envisaged to
efficiently process spatial data. It consists of three convolutional layers with 50, 25, and 10 kernels, each using ReLU activation
functions (Fig. 1 for more details). After the final convolutional operation, the resulting feature maps are flattened into a one-
dimensional vector, which is subsequently passed through a fully connected dense layer comprising 12,800 neurons, reshaping
to match the dimensions of the predictand.

### 2.4.3 U-Net

The U-Net model adopts an encoder-decoder architecture designed for spatial data processing. The encoder progressively ex-
tracts hierarchical features through a series of Convolutional Blocks (ConvBlocks), each containing convolutional layers with
ReLU activation, followed by $2\times2$ max pooling operations that reduce spatial dimensions while increasing the number of fea-
ture maps. The encoder expands from 64 to 512 channels, while reducing the spatial resolution from (80,160) to (10,20). In the
decoder, the model reconstructs the original resolution using $2\times2$ transposed convolutions (deconvolutions) that progressively
upsample feature maps (Prince, 2023). This is followed by convolutional layers that refine the output. The final layer consists
of a $1\times1$ convolution, followed by a linear activation function, producing an output of dimensions (1,80,160).



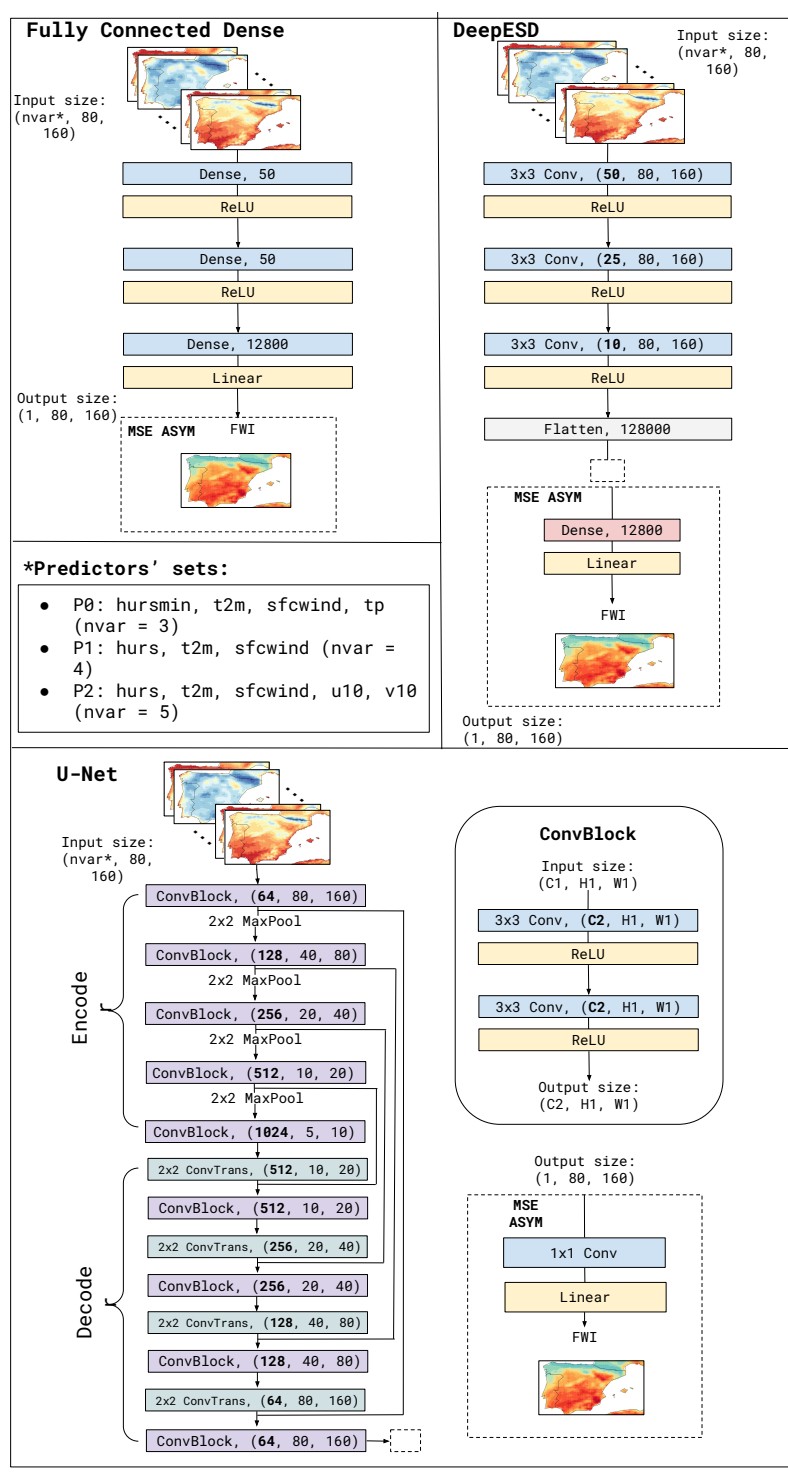

**Figure 1.** Schematic representation of the DL architectures used in this study, showcasing the different types of layers and their corresponding dimensions. The figure also highlights the distinct sets of predictors used as training inputs.





### 2.4.4 Loss functions

The DL models were fitted with two alternative loss functions: the mean squared error (MSE), which serves as a standard
loss function, and an asymmetric loss function (ASYM), designed to better capture extreme events, as introduced by Doury
et al. (2024). The MSE is defined as the average of the squared differences between observed and predicted values. This
metric penalizes larger errors more severely, making it useful for many regression tasks. However, MSE might not adequately
emphasize errors associated with extreme events. To address this limitation, we implemented an asymmetric loss function
defined as:

$$L_\theta = \frac{1}{N} \sum_{i=1}^{N} |y_i - \hat{y}_i| + \gamma^2 \times max(0, y_i - \hat{y}_i) \tag{1}$$

where $\gamma = G(y)$, where $G$ is the gamma cumulative distribution function (CDF) fitted to the time series of every grid point in
the training dataset. This loss function weights the mean absolute error (MAE) by an amount proportional to the extremity of
each value, thus assigning a higher penalty when the model underestimates extreme values. This tailored approach improves
the evaluation and prediction of extreme events in the dataset.

### 2.5 Model evaluation

In this study, the DL models are trained using daily data from the period 1979 to 2017. To evaluate the performance of the
model, 20% of these data is separated and treated as validation samples for model assessment during the training phase. The
final results are presented for an independent test period spanning 2018–2021. To assess the performance of the DL models and
the reproducibility of spatial and temporal patterns in the emulated results, several validation indices are used, each focusing
on different spatial and temporal aspects of the predicted series (Table 2).

| Metric | Description |
|---|---|
| MAE FWI | Mean Absolute Error for the FWI |
| MAE FWI95 | Mean Absolute Error for the FWI percentile 95-th |
| Freq. FWI95 Relative Bias | Relative bias of the frequency of extreme FWI events (over 95-th percentile) |
| Max Spell95 Bias | Inter-annual maximum spell length bias for extreme FWI events (over 95-th percentile) |
| ROC AUC | Receiver-operating characteristic (ROC) curve Area for extreme FWI events (over 95-th percentile) |
| $R^2$ | Coefficient of determination |

**Table 2.** Validation indices assessed in the study.

At each grid point, validation indices are computed, and their spatial distributions are depicted in the maps presented in
various figures in Section 3. These indices span different key characteristics of the predicted series to assess how models
emulate FWI, such as the mean FWI and extreme FWI distribution, or the length of spells for extreme FWI events. Instead of
using MAE (for FWI and FWI95) or bias (for Max Spell95), we assess relative bias with respect to the frequency of FWI95,





given by the following formula:

$$\text{Relative bias} = \frac{X - \hat{X}}{X} \times 100 \qquad (2)$$

where $X$ represents the observed and $\hat{X}$ the estimated value.

We also analyze the Receiver Operating Characteristic (ROC) curve, and the area Under the Curve (AUC), a graphical tool
for evaluating the performance of binary classification models (Hogan and Mason, 2011), in this case applied to assess the
DL model's ability to discriminate dangerous FWI events. It plots the true positive rate (sensitivity) against the false positive
rate at various decision thresholds and quantifies the overall discriminative ability of the model, with values ranging from
0.5 (random classification) to 1.0 (perfect classification). A higher AUC indicates better model performance in distinguishing
between positive and negative cases, making the ROC AUC particularly useful in unbalanced classification problems where
traditional accuracy measures may be misleading. In particular, the classification evaluated in the ROC AUC analysis pertains to
FWI events classified as 'extreme', defined as those exceeding the 95th percentile of the reference FWI, according to the danger
levels established by the Spanish Meteorological Agency (AEMET[1]), being thus impact-relevant for wildfire risk assessment
in the Iberian Peninsula. In addition to the validation indices listed in Table 2, we also use scatter plots the assessment of the
emulated FWI 95th percentile against reference FWI climatologies, where the $R^2$ coefficient of determination is used.

## 2.6 Model explainability

DL models have demonstrated remarkable predictive capabilities across various scientific fields, yet their complex architectures
often limit interpretability. This challenge has motivated the growth of eXplainable Artificial Intelligence (XAI), which aims
to provide tools for understanding the underlying relationships learned by machine learning models (McGovern et al., 2019).
In this work, we apply saliency-based methods to investigate the internal logic behind our DL predictions. Specifically, we
employ a variant of the Integrated Gradients (IG) technique (Sundararajan et al., 2017), which has been widely adopted in
climate applications (González-Abad et al., 2023; Kondylatos et al., 2022) for its effectiveness and ease of interpretability.

Traditional IG computes feature attributions by integrating the gradients of the model output with respect to the input, along
a linear path from a baseline input to the actual input. The method is formally expressed as:

$$s_i(x_i; x_b) = (x_i - x_b) \int_0^1 \frac{\partial f\left(x_b + t(x_i - x_b)\right)}{\partial x_i} \, dt \qquad (3)$$

where $x_i$ is the feature value, $x_b$ is the baseline (typically zero), and $f$ is the output of the model.

However, in this study we adopt a simplified approximation of IGs, where the integral is omitted and the feature relevance
is estimated directly from the gradients at the input point. This results in a computationally simpler, yet informative, saliency
map representation. The relevance of each input feature is therefore computed as:

---

[1]https://www.aemet.es/documentos/es/datos_abiertos/Estadisticas/IM_riesgo_incendios/eimri_generalidades.pdf



$$s_i(x_i) = x_i \cdot \frac{\partial f(x)}{\partial x_i} \tag{4}$$

This approach preserves the core idea of attributing the prediction to the input features based on the local sensitivity of the model, while avoiding the computational overhead of path integration. Although this approximation does not fully satisfy all axioms of the original IG method (Sundararajan et al., 2017), it has been shown to yield stable and interpretable attributions in practical applications (Toms et al., 2021).

To address known XAI challenges (Mamalakis et al., 2023), the raw saliency maps are post-processed by: (1) taking the
absolute value to focus on magnitude only, (2) zeroing all values below 10% of the per-sample maximum to suppress gradient noise (Toms et al., 2021), and (3) normalizing each map so its values sum to one, thus yielding relative contributions. These normalized maps are used for all subsequent explainability analyses.

This methodology enables us to quantify the relative importance of each predictor variable in the model's decision-making process, providing valuable insights into the learned relationships and the adequacy of the different sets of predictors tested,
aiding in the underlying physical interpretation of the results.

## 3    Results and Discussion

Several DL models are evaluated through an intercomparison of different predictor sets (Table 1) and the loss functions optimized during DL model training (Sec. 2.4.4). The results presented in this section focus on the main fire season over the Iberian Peninsula (June to September, JJAS; see e.g. Bedia et al., 2014) for the test period 2018-2021.

### 3.1    Proxy FWI validation

Figure 3 presents a multi-panel visualization, where the rows correspond to the validation measures shown in Table 2. From left to right, the first two columns represent the reference FWI and the Proxy FWI climatologies (P0, Table 1). The last column indicates the MAE or bias between both –depending on the evaluated index, Table 2–. The FWI95 frequency for the reference FWI is not shown, as it serves as the observed reference for error calculation: The 95th percentile of reference FWI is used as
baseline to compute the frequency of proxy FWI events exceeding this threshold (as well as emulated FWI in the following section).

Focusing on the differences between the Proxy and the reference FWI, we note that there exists greater MAE in Southern Portugal, North-western Spain and North-Eastern Spain, along the Ebro Valley. Conversely, the lowest MAE values (close to zero) are found along the Cantabrian and Mediterranean coasts, northwestern Portugal, the Pyrenees, and the Balearic Islands.
These low-MAE areas generally align with the lowest climatological FWI values (as depicted in first column, first row), except for the Balearic Islands, where climatological JJAS FWI remains relatively high (FWI≈35).

Regarding extreme FWI events (above reference FWI 95th percentile, FWI95), the Proxy FWI95 exhibits the highest MAE in the central-eastern and southeastern continental regions, coincident with very high reference FWI95 climatological values.



Although the Proxy FWI95 accurately represents southern Portugal, revealing low MAE values, overall, the most critical

FWI95 regions (mostly over Coastal Mediterranean and Central and Iberian Massifs) tend to be underrepresented by the Proxy FWI.

In regard with the temporal sequence of extreme event occurrence, the relative bias for the FWI95 frequency displays a spatial pattern similar to the Max Spell95, showing an overestimation of FWI95 events in Portugal, North-western Spain (Galicia) and also in the Cantabrian mountain range, the Pyrenees and the Ebro Valley, with a substantial underestimation in

the rest of the Iberian peninsula. Regarding the spatial distribution of Max Spell95, the Proxy FWI overestimates spell lengths in Galicia and Portugal, with an excess of up to four days. Conversely, in the continental Iberian Peninsula, the proxy FWI tends to underestimates spell length, remaining accurate over most of South-western Spain.





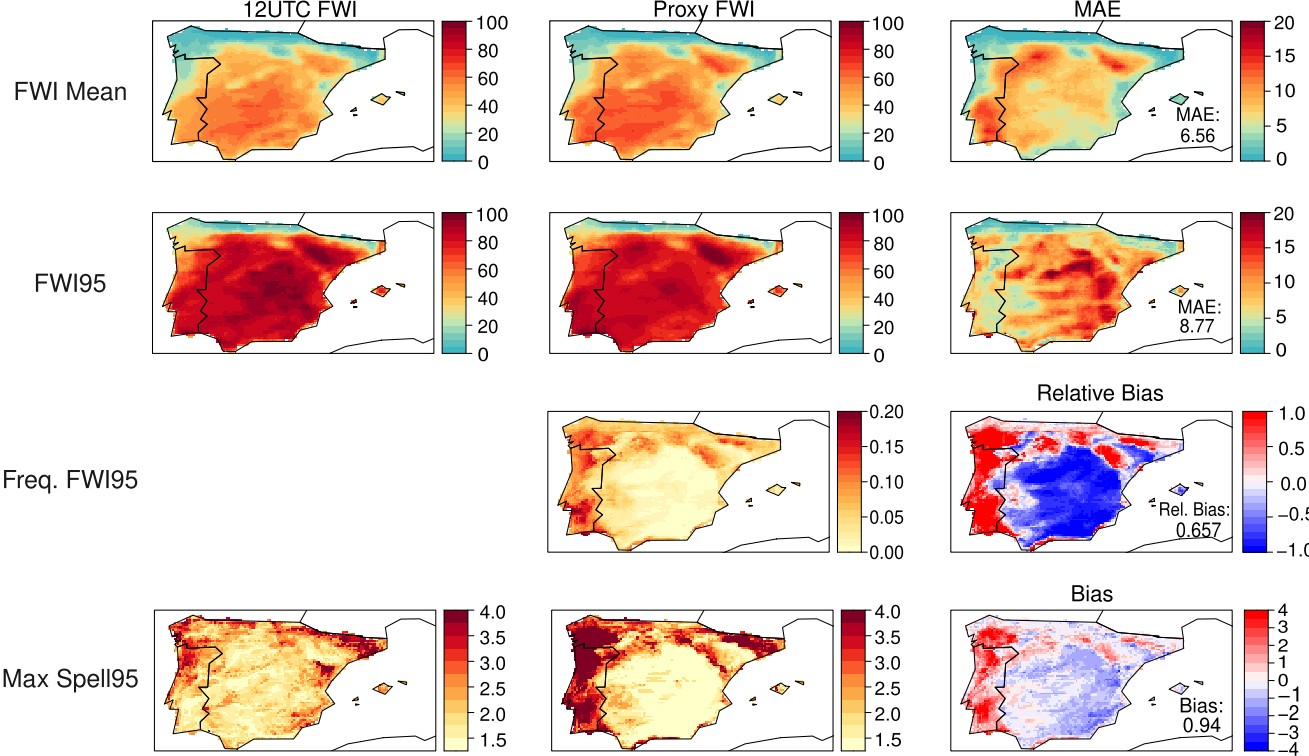

**Figure 2.** Reference and Proxy Fire Weather Index (FWI) climatologies for the fire season (June–September) during the test period (2018–2021). These include mean FWI, 95th percentile of FWI (FWI95), FWI95 frequency (number of days exceeding the reference FWI95) and the mean maximum annual spell (the number of days exceeding reference FWI95 Max Spell95). The third column displays the mean absolute error (MAE) for the FWI Mean and FWI95, the bias for Max Spell95 and the relative bias for the FWI95 frequency. The spatial averaged values of MAE or absolute (relative) bias are displayed at the bottom right.

## 3.2 Emulated FWI validation

This section presents an intercomparison of the different DL models tested, by evaluating errors relative to the reference FWI and comparing them with the Proxy FWI results described in the previous section, which serves as a benchmark. For brevity, here we present the results using the predictor set P0 as input (Table 1), considering that the overall intercomparison results are consistent regardless of the predictor set tested. Since P0 includes the variables necessary for computing the Proxy FWI using the standard FWI formulation, our goal is to assess whether different DL architectures are able to capture the reference FWI more accurately using the same input information.





### 3.2.1 Prediction errors

The prediction errors for the Dense, DeepESD, and U-Net models relative to the reference FWI are presented in Fig. 3. The columns correspond to the results of each DL model, and the rows display various validation metrics, including the mean absolute error (MAE) for FWI and FWI95 or the relative bias in the frequency of FWI95. Below the error maps, a scatter plot is displayed per DL model, including the proxy FWI, comparing the predicted climatological values against the reference FWI95.

Overall, all DL models exhibit similar performance across the validation indices compared to the Proxy FWI output. For each validation index, at least one DL model outperforms the Proxy FWI. While Dense and U-Net reveal slightly worst mean FWI MAEs (Fig. 3) than the proxy FWI (Fig. 2), DeepESD has a comparable performance (6.54 *vs.* 6.56). However, the DL models provide a smoother spatial distribution of MAE across both continental and coastal Mediterranean regions. In contrast, as discussed in Sec. 2.2, the Proxy FWI exhibits large MAE values in FWI-prone areas, such as the Ebro Valley, the Cantabrian mountain range, and southern Portugal (Fig. 2), which are largely reduced by the DL emulated FWI (Fig. 3).

Regarding FWI95 MAE, the U-Net model significantly improves upon the Proxy FWI results (7.03 *vs.* 8.77), offering a smoother MAE distribution across the region and particularly lower MAE values in southern Spain, where the Proxy FWI exhibits large errors. Except for Portugal and the Cantabrian coast, the U-Net model outperforms the Proxy FWI across Iberia. Meanwhile, the other DL models display considerable MAE values, especially in central and northeastern Portugal and the Cantabrian mountain range, where these errors are larger that those of the FWI proxy.

For FWI95 frequency, all three DL models outperform the Proxy FWI, with the Dense model achieving the lowest relative bias. The Dense and DeepESD models generally underestimate FWI95 frequency across the region, except in southeastern Iberia and some localized areas in northern Iberia. In contrast, the U-Net model performs worse than the other DL models, consistently overestimating FWI95 frequency across the entire domain.

To assess the temporal characteristics of the emulated FWI, we evaluate Max Spell95 w.r.t. reference FWI. Here, the U-Net model achieves the best average result, closely followed by DeepESD (bias of 0.68 *vs.* 0.69). The key difference between these models is that U-Net generally overestimates spell duration, with maximum exceedances of 1–2 days, whereas DeepESD tends to underestimate, with discrepancies of up to 2–3 days in specific areas across the Iberian Peninsula.

Despite these local problems, overall the scatter plots depict that DL models consistently improve upon the Proxy FWI results, regardless of the specific DL model. In terms of $R^2$, the U-Net model achieves the best performance for FWI95, with a value of 0.989, compared to 0.934 for the Proxy FWI. Overall, in the FWI95 scatter plots, the Proxy FWI underestimates higher values (above 60 for the reference FWI95), while the U-Net model, in particular, offers a significantly better representation of reference FWI.





**Figure 3.** DL model results for various spatial validation indices presented in Fig. 2. The results depict the differences relative to the reference FWI for the fire season (June–September) during the test period (2018–2021). The scatter plots compare climatology values between the Proxy FWI95 (in grey) and the corresponding DL model predictions (in blue), against reference FWI95. Linear fits are shown in grey for the Proxy FWI95 and in red for the DL model, with the corresponding $R^2$ values displayed in the top-left corner.

Furthermore in Figure 4, we assess the DL model ability in discriminating extreme fire danger events (i.e., reference FWI values above the 95th percentile) using ROC-AUC histograms, where the Proxy FWI is represented in red and the DL models in blue. Overall, the DL models improve upon the results provided by the Proxy FWI. Regarding extreme event detection,




DeepESD demonstrates the best performance, with a median ROC-AUC of 0.734, compared to 0.727 for U-Net and 0.7 for the Dense model, as shown in the histograms (Fig. 4). In this regard, the DL models outperform the Proxy FWI, as indicated by the higher frequency of ROC-AUC values above 0.5 (indicating better-than-random classification). The only exception occurs for ROC-AUC values above 0.9, which correspond to the classification of grid points along the Cantabrian coast, where climatological FWI is the lowest Iberia (Fig. 2) due to generally milder and moist conditions throughout the year, where proxy FWI exhibits better FWI95 event discrimination than DL models.

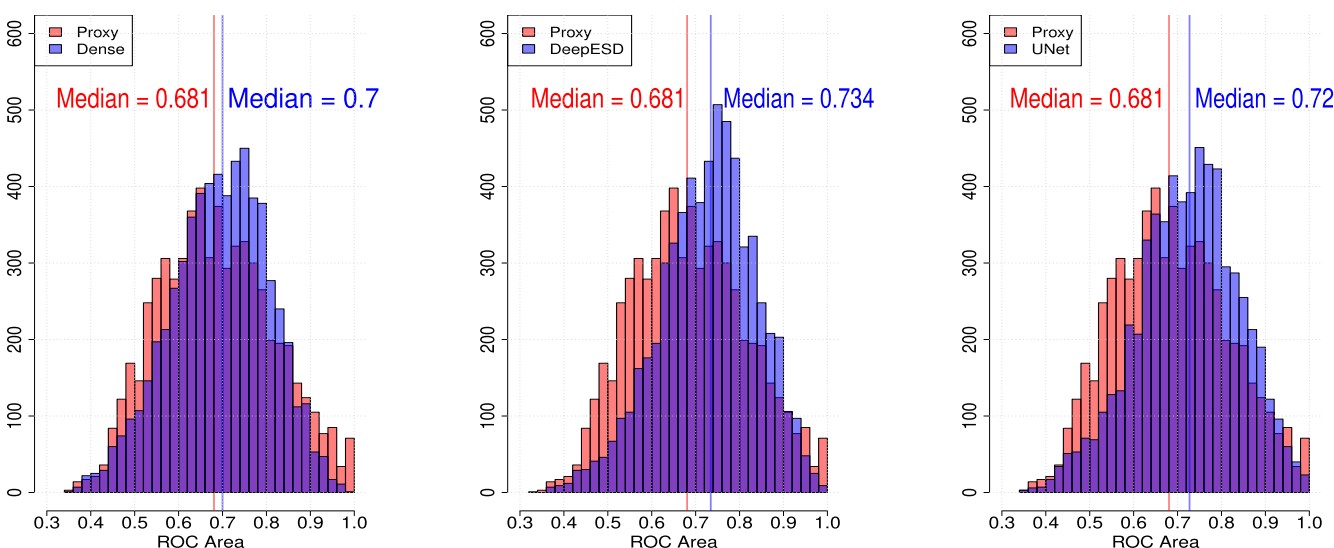

**Figure 4.** DL prediction results corresponding to the fire season (June–September) for the test period (2018–2021). Histograms compare the Area Under the ROC Curve (AUC)for extreme FWI event classification (above the 95th percentile) at each grid point, between the Proxy FWI (in red) and the corresponding DL model (in blue). The median values of both are also indicated.

### 3.3 Emulation Model Explainability

Here, we analyze the saliency attributed to the predictor variables, grouped by the same climatic regions and fire danger categories defined in Appendix A. The overall results indicate no major differences in the saliencies of the different DL models, underpinning their robustness and physical coherence. Therefore, here we present the explainability results of the U-Net model, since it outperforms proxy FWI and attains similar validation results as DeepESD, while offering greater computational efficiency. To this aim, we compute the saliencies of each variable considering the time events corresponding to the different fire danger categories separately (Fig. 5).

Focusing on the *high*, *very high*, and *extreme* fire danger categories, temperature and relative humidity emerge as the most influential variables in the model's predictions. The relative importance of these two variables varies across climatic regions. In the Continental Mediterranean region, temperature and relative humidity show similar levels of saliency. For the *high* fire danger category, relative humidity slightly surpasses temperature, whereas for the *very high* and *extreme* categories, temperature becomes more dominant. Wind speed consistently ranks as the third most important variable in these categories, although





its relevance diminishes as the fire danger increases. Precipitation shows the lowest saliency in all categories and becomes increasingly negligible for higher danger categories.

The Coastal Mediterranean region shows a similar pattern. However, in the *high* category, temperature becomes more salient than relative humidity. In contrast, for the *extreme* category, relative humidity overtakes temperature as the most relevant predictor.

Similarly, in the Atlantic region, temperature is the most significant variable for the *high* and *very high* categories. For the *extreme* category, however, relative humidity surpasses temperature. Notably, in this region, the difference in saliency between wind speed and the top-ranked variables is smaller compared to other regions, suggesting a higher reliance on this variable in this region. As with the other regions, the relevance of wind speed decreases with increasing fire danger, and precipitation remains consistently negligible.

For the *medium* and *low* fire danger categories, the importance of variables shifts more markedly and shows less consistency across climatic regions. In the Continental and Coastal Mediterranean regions, precipitation becomes the most relevant variable for the *low* category, followed by relative humidity, wind speed, and temperature. For the *medium* category, wind speed emerges as the most salient predictor, followed by temperature, relative humidity, and precipitation.

In contrast, in the Atlantic region, relative humidity is the most important variable for the *low* category, followed by precipitation, wind speed, and temperature. For the *medium* category, temperature takes the lead, followed by wind speed, relative humidity, and finally precipitation. These results suggest that precipitation is a key variable for *low* fire danger levels but loses significance as fire danger increases, becoming practically irrelevant in *extreme* conditions, when temperature and relative humidity gain prominence and become the most critical predictors. This is consistent with numerous studies that highlight the

importance of high temperatures and low humidity in contributing to extreme fire danger (Jain et al., 2022).

Given that precipitation has negligible saliency in high percentile fire danger cases, which are precisely the most important from the point of view of impacts, the next section evaluates model performance after removing precipitation from the predictor set. We also examine alternative predictor sets, including replacements for precipitation and wind speed, such as eastward and northward wind components, under the hypothesis that these may carry additional useful information for the emulators.




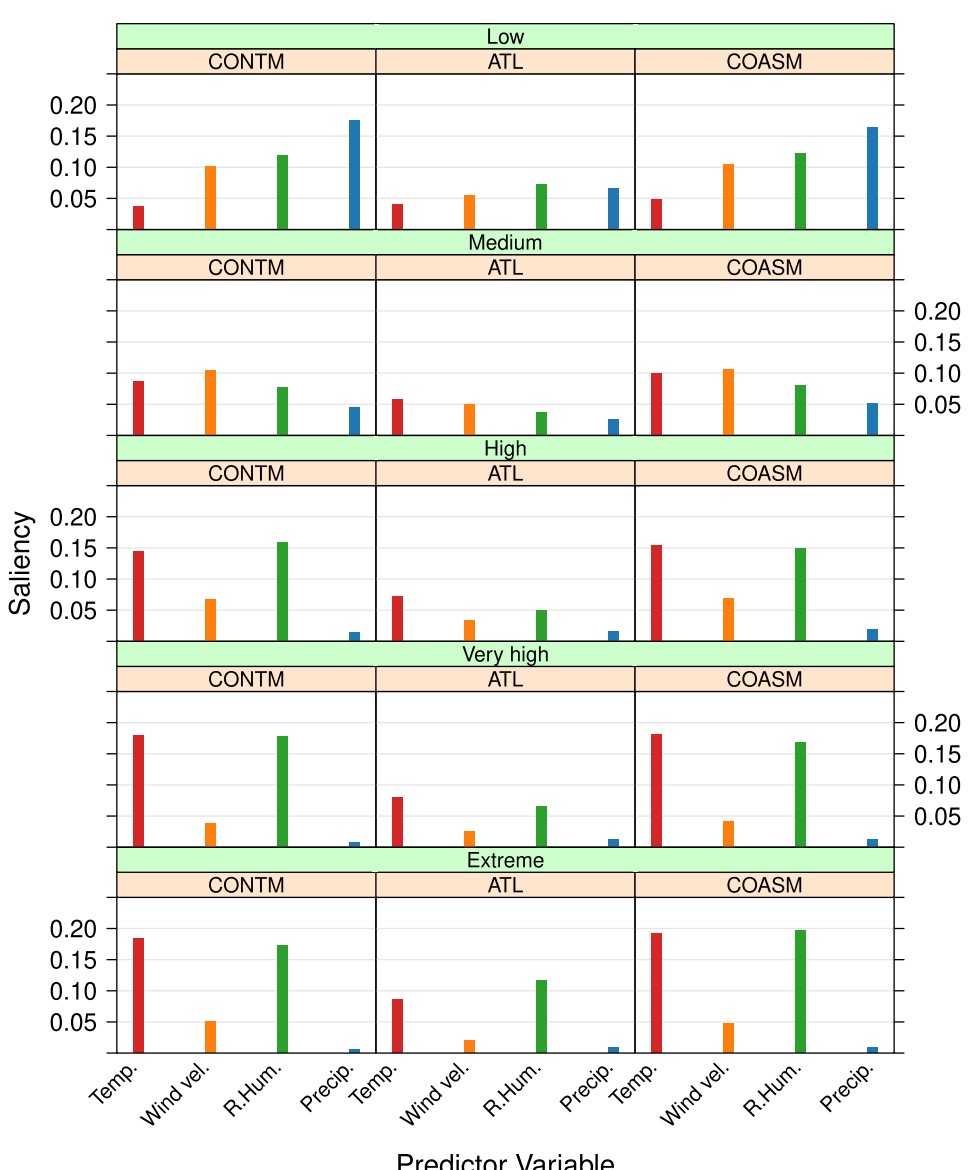

**Figure 5.** Aggregated saliency of predictor variables by climatic region and fire danger category for the U-Net model. Columns represent the climatic regions defined in Fig. A1, while rows correspond to the fire danger categories described in Section 3.2. The displayed saliency values are derived from saliency maps, normalized at each grid point. Results for the JJAS season, test period (2018-2021).



## 3.4 Predictor set intercomparison

In this section, we intercompare different alternative input predictor sets, summarized in Table 1. For simplicity, we focus on the U-Net DL method as in Sec. 3.3. The U-Net trained with P0 and P1 yields better results than when trained with P2, except for the FWI95 Frequency (Fig. 6, analogous to Fig. 3 for comparability). The U-Net P2 exhibits a larger MAE for the FWI across the entire domain, except along the Cantabrian coast. The difference between U-Net P0 and U-Net P1 is relatively small, with the most remarkable variations observed in the Continental and Mediterranean coastal regions. For the MAE of FWI95 events, U-Net P2 significantly overestimates across the entire domain, whereas U-Net P0 and U-Net P1 present similar MAE spatial distribution. However, U-Net P1 shows a greater presence of errors in specific regions, such as eastern Portugal, the Pyrenees, and the Ebro Valley. Regarding the frequency of FWI95 events, U-Net P0 tends to overestimation across nearly the entire domain, while U-Net P1 and U-Net P2 exhibit similar spatial patterns, with both models attaining comparable performance. Among these, U-Net P1 appears to be the most effective in capturing this index. Examining the Max Spell95 results of the different experiments, U-Net P2 performs the worst, with overestimation up to 3 to 4 days in some areas, whereas the largest differences in P0 and P1 range between 2 and 3 days at most. U-Net P1 shows a slight improvement over U-Net P0, displaying more areas of underestimation, while U-Net P0 predominantly overestimates across most of the domain. Finally, the FWI95 scatter plots exhibit a high $R^2$ fit for all three predictor sets, outperforming the Proxy FWI. In terms of overall performance, U-Net P0 produces the best results, followed by U-Net P1 and U-Net P2, though the differences among the latter remain relatively small.





**Figure 6.** UNet model results for various spatial validation indices presented in Fig. 2 and different predictors sets assessed (see Table 1). The results depict the differences relative to reference FWI for the fire season (June–September) during the test period (2018–2021). Additionally, scatter plots compare climatology values of the Proxy FWI95 (in grey) and the UNet model (in blue) against reference FWI95. Linear regression fit lines are shown in grey for the Proxy FWI95 and in red for the UNet model (to enhance visualization), and the corresponding $R^2$ values displayed in the top-left corner.

Overall, in Figure 7, U-Net P1 yields the best results as classifiers for extreme events. These experiments improve upon the median results of the Proxy FWI, with U-Net P1 emerging as the most effective model in this context.





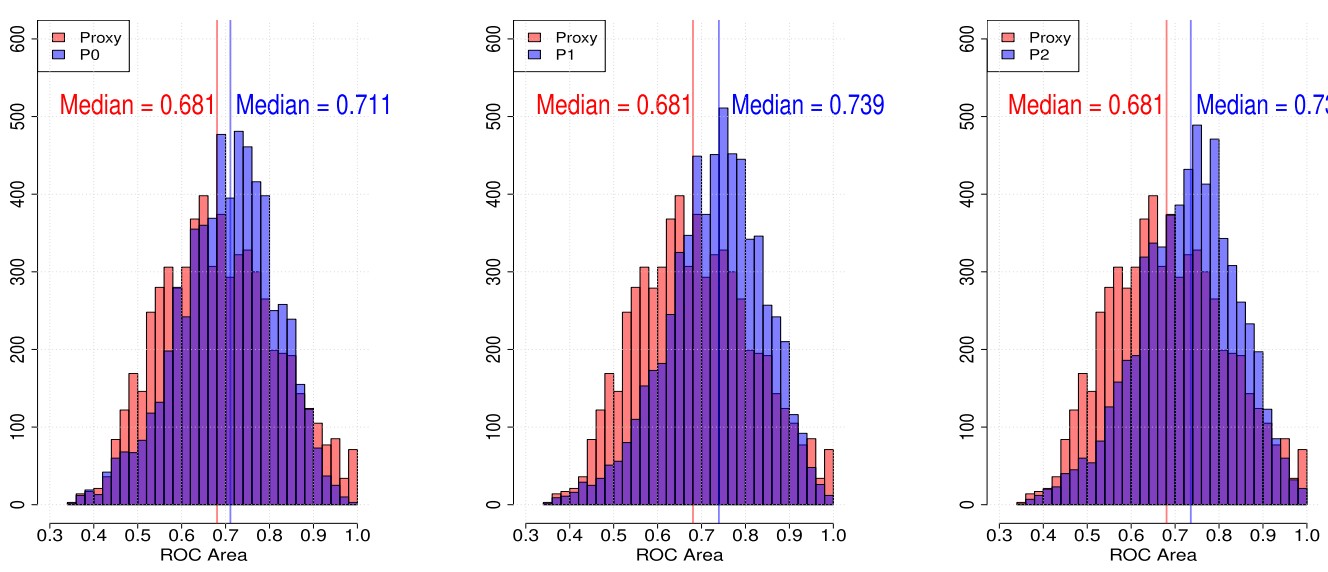

**Figure 7.** UNet model histograms resultus for the fire season (June–September) for the test period (2018–2021). Histograms compare the Area Under the Curve (AUC) of the Receiver Operating Characteristic (ROC) curve for extreme fire danger classification (above the 95th percentile) at each grid point between the Proxy FWI (in red) and the UNet model (in blue). The median value is indicated for both results.

In summary, the U-Net model demonstrates superior performance over the Proxy FWI, with U-Net P1 generally excelling in overall performance. The inclusion of wind components as additional predictors in P2 has a deleterious effect on the temporal and distributional similarity of emulated and reference FWI, suggesting that despite the selective use of features internally undergone by DL methods, a previous predictor screening might be necessary to ensure optimal results.

### 3.5 Loss function assessment

We next present a summary of the validation results, including an intercomparison of alternative MSE and asymmetric loss functions in the DL models (Sec. 2.4.4). Table 3 presents the validation indices for different predictor sets and DL models, comparing their performance against the Proxy FWI baseline, where the ASYM loss function values are indicated in parentheses. For all indices, the DL models exhibit consistent improvements over the Proxy FWI, with at least one model outperforming it in each category.

Regarding the loss function, the ASYM loss generally improves performance in most validation indices. The ASYM-trained models tend to yield lower MAE values, particularly for DeepESD in P0 (6.81 vs. 6.54) and P2 (7.12 vs. 6.24), showing better accuracy in estimating FWI. Similarly, in the FWI95 MAE, the ASYM-trained models consistently outperform MSE loss counterparts, as seen with DeepESD in P0 (9.28 vs. 8.62) and P2 (9.66 vs. 9.44). In terms of extreme event prediction, the Max Spell95 Bias shows inconclusive results. While the ASYM loss function improves the bias in some cases, such as U-Net in P2 (0.93 vs. 0.54), in other cases, it worsens bias, as observed with Dense in P2 (0.78 vs. 1.05). This suggests that ASYM does not consistently improve the temporal persistence of extreme events but may still provide benefits in specific cases. For





the Relative Bias Frequency of FWI95, ASYM provides substantial benefits in reducing bias. The most notable case is U-Net in P1, where the ASYM-trained model achieves a significantly lower bias (0.383 vs. 0.269), reinforcing ASYM's ability to enhance the reliability of extreme event frequency estimations.

Overall, ASYM appears to be particularly beneficial in reducing error in FWI estimations (MAE and MAE FWI95) and in improving the classification of extreme fire weather danger, as evidenced by the lower relative bias for FWI95. However, its benefits on temporal aspects, such as Max Spell95 Bias, are not consistent. These findings suggest that while ASYM is generally advantageous, its effects are model- and index-dependent, and its selection should be carefully considered depending on the primary objective of the analysis.

| Predictors sets | DL Model | MAE | MAE FWI95 | Max. Spell95 Bias | Rel. Bias Freq. FWI95 |
|---|---|---|---|---|---|
| Proxy FWI | | 6.56 | 8.77 | 0.94 | 0.657 |
| P0 | Dense | 7.01 (7.94) | 10.72 (8.86) | 0.73 (0.95) | 0.337 (0.662) |
| | DeepESD | 6.54 (6.81) | 9.28 (**6.82**) | 0.69 (0.6) | 0.357 (0.401) |
| | UNet | 7.04 (**6.09**) | 7.03 (8.51) | 0.68 (**0.65**) | 0.561 (**0.313**) |
| P1 | Dense | 6.85 (7.33) | 10.97 (9.83) | 0.7 (0.92) | 0.342 (0.514) |
| | DeepESD | 6.74 (6.43) | 8.43 (8.01) | 1 (0.74) | 0.666 (0.46) |
| | UNet | 6.38 (**5.84**) | **7.97** (8.46) | **0.63** (0.63) | 0.383 (**0.269**) |
| P2 | Dense | 6.95 (7.58) | 10.72 (9.8) | 0.78 (1.05) | 0.408 (0.588) |
| | DeepESD | **6.24** (7.12) | 9.66 (9.44) | 0.69 (0.95) | 0.362 (0.521) |
| | UNet | 8.4 (6.95) | 9.57 (**8.7**) | 0.93 (**0.54**) | 0.552 (**0.295**) |

**Table 3.** Validation results of DL FWI emulators for different predictor sets. Values in parentheses are for the ASYM loss function for model training (default values correspond to MSE loss). The first row presents the Proxy FWI results, as benchmark. Bold values highlight the best validation index within each predictor set, while underlined values denote the overall best across all indices.

# 4 Conclusions

In this study, we have shown that DL emulators can effectively represent actual FWI using input variables of different temporal resolution and/or daily aggregation function as input. We have shown this for different DL methods and different sets of inputs using readily available ERA5-Land reanalysis predictors. Our analysis indicates that DL models not only capture the complex spatial and temporal characteristics of the reference FWI with higher fidelity than traditional proxy-based approaches, but also improve the detection of most dangerous situations allowing for a more accurate classification of extreme fire danger events. Our results strongly suggest that employing daily mean inputs, as opposed to instantaneous data, ensures adequate accuracy for operational FWI emulation. This finding is particularly valuable because daily datasets are more commonly accessible in climate repositories, thus lowering the barrier for integrating FWI predictions into fire management and climate risk assessments.



Furthermore, our investigation of different predictor sets revealed that excluding certain variables, such as precipitation, does not significantly degrade model performance. This suggests that simpler input configurations can be leveraged to achieve robust predictions, thereby reducing computational overhead and simplifying model implementation.

The XAI analysis reveals that temperature and relative humidity are the most influential predictors in high fire danger conditions, with their relative importance varying by climate regions. Wind speed holds moderate relevance but diminishes as fire danger intensifies, while precipitation consistently ranks as the least important variable and becomes nearly irrelevant under extreme fire danger conditions. In low and medium fire danger categories, the importance of variables has a greater regional variability, with precipitation emerging as a key factor in some cases. These findings underscore the critical role of temperature and relative humidity in high-to-extreme fire danger predictions and justify the consideration of alternative predictor sets that exclude precipitation as a non-essential predictor for dangerous event prediction.

The implications of our work extend beyond the immediate context of FWI estimation, and sets the stage for the emulation of multivariate climate indices relying on meteorological variables that are not commonly found in observational records or difficult to obtain from most climate model output repositories, given a sufficiently comprehensive database for its training (e.g., a high resolution reanalysis). By showcasing that DL models can serve as efficient surrogates for complex, computationally intensive fire danger indices, our study opens up new avenues for integrating AI into operational forecasting systems. This approach can significantly expedite the decision-making process in fire-prone regions, where timely and accurate risk assessments are critical.

Looking ahead, future research should focus on refining these models further by incorporating additional meteorological and environmental predictors, and by testing their transferability across different climatic regions. Ultimately, the continued development and operationalization of DL-based emulators have the potential to transform wildfire risk management, offering more precise, interpretable, and accessible tools for both researchers and practitioners.

*Code and data availability.* An illustrative example providing reproducible code and data via jupyter-notebook is provided in the open GitHub repository https://github.com/SantanderMetGroup/DeepFWI, where access to the required open data curated in Zenodo is granted and software environment configuration is detailed. Please note that due to brevity and the significant computing infrastructure required for full analysis reproducibility, the examples provided are a small sample of function calls and model configurations. These examples are applied on a coarser-than-native ERA5-Land grid for a limited time period, making them suitable for running locally on a CPU and avoiding extensive computing times. Further details or complete training/test datasets are available upon request to the authors.

## Appendix A:  Climatological regions

Here, we present the division of the Iberian Peninsula into three regions—Atlantic (ATL), Continental Mediterranean (CONTM), and Coastal Mediterranean (COASM)—for the eXplainable Artificial Intelligence (XAI) analysis provided in the main manuscript (Figure 5).



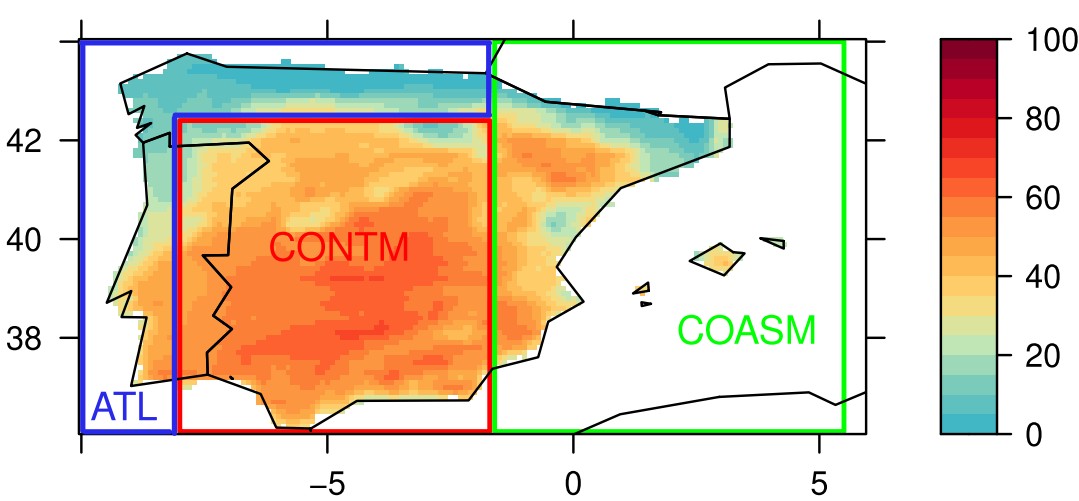

**Figure A1.** Climatological regions within the Iberian Peninsula used for XAI analysis in Fig. 5. Blue, green and red rectangles indicate ATL (Atlantic), COASM (Coastal Mediterranean), and CONTM (Continental Mediterranean) regions respectively.

*Author contributions.* O.M., J.B. and J.B-M. were responsible for the development and implementation of the code used in the experiments. O.M, J.B., J.M.G and J.B-M. contributed to the experimental framework design. All authors contributed to the analysis and interpretation of the results. All authors contributed to manuscript writing and they reviewed and approved the final manuscript.

*Competing interests.* The authors declare that no competing interests are present.

*Acknowledgements.* We thank our colleague, Dr. Jose Abad, for his insightful and fruitful discussions on deep learning model con-
figuration and explainability approaches. O.M. has received research support from grant PRE2021-100292 funded by MCIN/AEI /10.13039/501100011033, as part of the R+D+i project CORDyS (PID2020-116595RB-I00) with funding from the Spanish Ministry of Science MCIN/AEI.10.13039/501100011033. J.M.G. and J.B. acknowledge funding by the Ministry for the Ecological Transition and the Demographic Challenge (MITECO) and the European Commission NextGenerationEU (Regulation EU 2020/2094), through CSIC's Interdisciplinary Thematic Platform Clima (PTI-Clima). J.B. has received research support from Grant PID2023-149997OA-
I00 (PROTECT Project) funded by MICIU/AEI/10.13039/501100011033 and by ERDF/EU. P.M.M.S. acknowledges project DHEFEUS (https://doi.org/10.54499/2022.09185.PTDC) and UID/50019/2025 and LA/P/0068/2020 https://doi.org/10.54499/LA/P/0068/2020.



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
