# Peer review of "Deep Learning Emulation of Multivariate Climate Indices: A Case Study of the Fire Weather Index in the Iberian Peninsula"

_EGUsphere, 2025_

## Author Comment (AC1)

We sincerely thank the reviewer for taking the time to review our manuscript and for the constructive comments. We believe their feedback has improved the clarity of the manuscript and overall quality of this work.

**I'm a bit confused as to how you have chosen the predictor sets for different experiments, namely P1 and P2. It does not seem to have been done systematically , at least how it is described. For eg. in L115-L120 you say that you have done experiments to avoid precipitation due to it's complex nature, which in itself is fine. But you also say that P1 and P2 are done to improve upon the result of the best proxy P0 which uses DM Temp, Min Rel. Hum., daily accum precip and DM wind speed. The following questions arise based mainly from the way Table 1 is presented:**

1. **Why is Min Rel Hum. changed to DM Rel. Hum for P1 and P2 along with removing precip? Were other experiments done with precip included?**
2. **Have you done an experiment with the following as predictors: daily mean Temp, min. Rel. Humidity, and daily mean wind speed?**

In this study, the Deep Learning (DL) models are compared against the optimal Proxy FWI version from Bedia et al. (2014), which serves as our benchmark. This benchmark estimates the daily FWI based on a selection of atmospheric variables: daily mean temperature, minimum relative humidity, daily mean wind speed and 24-hour accumulated precipitation. To ensure the fairest possible comparison between methods, we defined a predictor set for the DL models which contains the latter list of variables (P0).

The subsequent shift from minimum to daily mean relative humidity in predictor sets P1 and P2 is intentional, driven by data availability. Our objective with these new predictor sets was to rely exclusively on daily mean variables, as these are more consistently and widely available in climate model outputs and reanalysis datasets than variables such as minimum relative humidity.

Precipitation presents an additional challenge, as it is a complex variable that climate models often struggle to reproduce. For this reason, we investigated whether robust performance could still be achieved in the absence of precipitation, by excluding it from predictor sets P1 and P2. This design choice also makes the DL models less sensitive to uncertainties in precipitation estimates (L115–L120 in the manuscript). Predictor set P0, which includes daily accumulated precipitation, therefore serves as our baseline configuration. Importantly, our explainability analysis (Section 3.3) confirms that precipitation contributes only minimally to predicting extreme FWI values (e.g., FWI95), reinforcing the notion that the models can maintain strong performance even without it regarding FWI extremes.

Following the reviewer´s question, we have conducted a dedicated experiment using only daily mean temperature, minimum relative humidity, and daily mean wind speed. In this case, we compare P1 with P1 using minimum relative humidity instead of daily mean relative humidity (first and third column of Figure 1). Figure 1 presents the climatology results as is

shown in several Figures in the manuscript: the first row shows the mean FWI, the second row shows the FWI95, the third row indicates the Relative Bias of the FWI95 Frequence and the Maximum Spell95 Bias is presented in the fourth row. As shown, the results obtained with minimum relative humidity are worse in terms of FWI95 MAE than those using mean relative humidity, but similar in terms of mean absolute error (MAE). Moreover, according to the referee suggestions we have tested a U-Net considering the P1 adding precipitation. The results of this aforementioned pattern are competitive with respect to the P1 pattern presented in the manuscript. For instance, P1 + precipitation pattern improves the performance in estimating some of the validation indices assessed. However, we observe that the results for the FWI95 MAE are worse than using the pattern P1. This is linked with the discussion of xAI results in the Section 3.3 of the manuscript. Including precipitation can model better the whole distribution of the FWI since precipitation might be linked to low and moderate FWI events. However, in the case of extremes (FWI95) there is a lack of importance of this variables in the extremes prediction. That may be the reason why U-Net trained with P1 plus precipitation get competitive results in the MAE FWI index, but on the contrary in FWI95 get worse results than the FWI95.

These additional tests have been mentioned in the manuscript in Section 2.3: "*Other predictors sets have been tested, such as P1 using minimum relative humidity instead of daily mean relative humidity or P0 including the lagged 24-hour precipitation, without any added value compared to the configurations mentioned in the manuscript. These experiment results are shown in Figure D2.*" *Moreover, a Figure (Figure D2) has been included in the new Appendix D: Appendix D: Sensitivity analysis of deep learning models and evaluation of other predictors sets.*

[Figure]

Figure 1: *From left to right, results from the U-Net model trained with the P1 predictors set (see Table 1 from manuscript), trained with P0 and adding the 24-h lagged precipitation,*

*trained with P1 adding minimum relative humidity instead of daily mean relative humidity and trained with P1 adding precipitation. The maps display differences relative to the reference FWI for the fire season (June–September) during the test period (2018–2021) for different validation indices. MAE represents the spatially aggregated mean absolute error of the deep learning predictions with respect to the FWI reference, while Bias denotes the spatially averaged bias in absolute value.*

*Bedia, J., Herrera, S., Camia, A., Moreno, J. M., and Gutierrez, J. M.: Forest Fire Danger Projections in the Mediterranean using ENSEMBLES Regional Climate Change Scenarios, Climatic Change, 122, 185–199, https://doi.org/10.1007/s10584-013-1005-z, 2014*

**It may actually be only required to perhaps restructure L115-L120 to better suit Table 1 to avoid confusion.**

We acknowledge that lines L115–L120 in the original manuscript could create confusion for the reader regarding the motivations behind the predictor configurations. In particular, the sentence "[P1 and P2 are evaluated as alternative emulation inputs (Table 1), aiming to improve upon P0 results.]" was unclear and potentially misleading. To address this issue, we have revised the paragraph in the updated version of the manuscript to clarify our rationale and improve readability.

"To emulate the reference FWI, we consider several predictor sets derived from ERA5-Land, summarized in Table 1. The initial experiments use P0, which builds on the same set of variables used in the optimal FWI approach from Bedia et al. 2014 (hereafter Proxy FWI). This allows us to assess whether the DL models can more accurately replicate the FWI transfer function, when provided with the same predictors as the Proxy FWI. Accordingly, P0 includes 24-hour accumulated precipitation, daily mean air temperature and wind speed, and minimum relative humidity. Proxy FWI is derived from the standard FWI formulation using the latter set of variables.

The use of minimum relative humidity in P0 ensures the fairest possible comparison with Proxy FWI. However, in predictor sets P1 and P2, we intentionally replace minimum relative humidity with daily mean relative humidity. This is motivated by the fact that daily mean variables are more consistently provided by climate model outputs, and we wanted to examine the performance of the emulator under such predictor-limited cases.

Precipitation poses an additional challenge: it is one of the most difficult variables for climate models and reanalyses to represent reliably, due to its strong spatial and temporal variability, its dependence on complex physical processes, and the influence of multiple interacting factors. To examine whether robust performance can still be achieved in its absence, we exclude precipitation from predictor sets P1 and P2.

Finally, in P2 we tried measuring the impact of using wind speed module versus wind speed zonal components, which could be relevant when determining fire danger."

**L155-L165: How do you make sure that the model doesn't overestimate low-moderate values to compensate for extreme value underestimation? Was this tested at all? I**

**understand that it may not be relevant if you're only interested in extreme cases. However, the total performance of the model still needs to be physically consistent.**

*Thank you for your comment. The ASYM loss function is specifically designed to better capture extreme events, particularly in the right tail of the distribution. We have verified that models trained with this loss function do not overestimate low to moderate values in the case of the U-Net model. However, for the other models (Dense and DeepESD), we observe an overestimation in the lower and moderate ranges. A graphical comparison is provided below, showing the distribution of predictions using QQ plots for a deep learning model trained with both the MSE and ASYM loss functions.*

*QQ plots were constructed to assess the agreement between observed FWI, proxy FWI, and deep learning model predictions. For each model, empirical quantiles of the observed FWI were computed at probability levels from 0 to 0.99 in steps of 0.01. The same quantiles were calculated for both the proxy and model outputs, using the FWI daily values for all grid points concatenated after spatial subsetting and seasonal masking.*

 *The QQ plots were then obtained by plotting the observed FWI quantiles (reference distribution) against the corresponding quantiles of the proxy and the model. In addition, mean squared errors (MSE) between the observed FWI and proxy quantiles, as well as between the observed FWI and model quantiles, were computed to quantify discrepancies. To further evaluate model performance in the upper tail of the distribution, the procedure was repeated for the 95th to 99.9th percentiles (in steps of 0.001), and the resulting QQ plots were shown as inset panels. Finally, threshold lines corresponding to key percentiles (40th, 65th, 85th, and 95th) of the observed distribution were included to highlight different fire risk categories.*

[Figure]

*the models architectures (Dense, DeepESD, U-Net). The solid blue line represents the model predictions, while the black dashed line indicates the 1:1 correspondence. Vertical dashed lines mark percentiles of the proxy distribution.*

*Figure 2 presents Q-Q plots comparing predicted and proxy values for different models trained with two different loss functions: MSE (first row) and ASYM (second row). A key inspection of these plots show us that the Dense and DeepESD models with ASYM as loss function overestimates low and moderate values of the distribution, a behavior which is not observed when training with the MSE. In the case of the U-Net model this overestimation is not observed, and indeed training with the MSE as loss function yields a slight overestimation of low and moderate values. This suggests that the ASYM loss function—while tailored to better capture extreme events—does not compromise the U-Net model performance in the lower quantiles, however for the Dense and DeepESD models low and moderate values for the FWI are less accurate than when considering the MSE as loss function. Therefore, the ASYM loss function can compromise the performance of low and moderate levels in some cases.  This  insight has been added to the manuscript in Section 3.5: "These findings suggest that while ASYM is generally advantageous for modeling the tail of the distribution, it can sometimes lead to an overestimation of low and moderate FWI values. While the ASYM loss function does not compromise the U-Net model performance in the lower quantiles, it leads to an overestimation for the Dense and DeepESD models of low and moderate values for the FWI as compared to results from models trained with MSE as loss function. Therefore, the ASYM loss function can deteriorate the performance of low and moderate levels in some cases, and its selection should be carefully considered depending on the primary objective of the analysis." Also we want to highlight that the QQ-Plot Figure is added as Figure C1 in the new appendix section C: QQ-plots assessment.*

**L220: Is this a description of Fig. 2?**

*Yes, line 220 refers to Figure 2. Thank you for highlighting this typo.*

**L256-L261: The authors may perhaps explain if the smoother outputs from DL models are good/bad? And perhaps also give an explanation why it's smooth.**

> 1. **Have you considered using some form of terrain-based predictor, say topography field for example? It might help in providing a more discernible profile to the fields.**
> 2. **Since you're estimating FWI, I would expect that vegetation cover would be a necessary predictor. Why was it not used?**

*We thank the referee for the thoughtful observations. Regarding the smoothness of the DL model outputs, we acknowledge that this is a common feature of neural network-based emulators, particularly when trained to minimize mean squared error. In our view, the smoother fields are not necessarily detrimental; rather, they reflect the models' tendency to average out local noise and variability that may not be strongly represented in the training data. In operational contexts, smoother outputs may even be advantageous, as they reduce false alarms and improve spatial coherence. Nonetheless, we agree that further work could*

*explore methods to enhance spatial detail, such as Super-Resolution Techniques, spatial attention mechanisms or other post-processing techniques.*

*Concerning the use of terrain-based predictors, we appreciate the suggestion. While topography can influence fire behavior and fuel moisture indirectly, the Fire Weather Index (FWI) is designed to quantify meteorological fire danger independently of terrain or vegetation characteristics. It is computed solely from weather variables (temperature, humidity, wind, and precipitation), and does not include any static environmental inputs. Therefore, introducing terrain predictors would not be consistent with the formulation of the reference index we aim to emulate.*

*Similarly, vegetation cover is not used in the computation of the FWI. The index is not a fire risk or impact model, but rather a meteorologically driven indicator of fire danger potential. It assumes a generic fuel type and does not account for land cover, fuel load, or vegetation type. For applications where vegetation is relevant—such as fire spread modeling or impact assessment—additional predictors would indeed be necessary. However, in the context of this study, which focuses on emulating the FWI as defined by its original formulation, the inclusion of vegetation data is not required.*

**L300-L302: If it is not too much extra work, I would like to see how previous days' (e.g. 24h prior) precipitation (lagged precipitation) might affect the model capabilities? The simplest model would suffice. If it is not possible to run the models, then you can also explain how this would affect/not affect the models' performance (with necessary refs).**

*We appreciate the suggestion to include lagged precipitation as an additional predictor. To evaluate its impact, we trained a version of the U-Net model using the P0 set (surface air temperature, minimum relative humidity and surface wind speed), adding 24-hour lagged precipitation as an extra input.*

*As shown in the result , the inclusion of lagged precipitation in the predictor set leads to noticeable changes in model performance (see Figure 1 of this document, columns 1-2). While it helps to reduce certain biases, this improvement comes at the cost of higher errors in the upper tail of the FWI distribution, particularly for extreme values. This trade-off suggests that lagged precipitation does not consistently enhance the model's ability to capture critical fire danger conditions, which are most relevant for our analysis.*

*These results are consistent with the saliency analysis presented in Section 3.3 (Figure 5 in the manuscript). As shown in the manuscript, precipitation gains importance when predicting low to moderate FWI values; however, for extreme FWI levels (high, very high, or extreme), the contribution of precipitation is negligible. This could explain why the MAE across the full FWI distribution is lower when lagged precipitation is included as an input, while performance for FWI95 worsens under the same setting. Since our main focus is on*

*accurately emulating extreme FWI values in the right tail of the distribution, we consider that including lagged precipitation in the predictor set is not the most suitable approach. These additional tests have been mentioned in the manuscript in Section 2.3: "Other predictors sets have been tested, such as P1 using minimum relative humidity instead of daily mean relative humidity or P0 including the lagged 24-hour precipitation, without added value compared to the configurations mentioned in the manuscript. Therefore, these experiments are not included in the text for simplicity"*

*It is also important to note that the FWI system itself embeds memory of past wet and dry conditions. Its subcomponents, particularly the Drought Code (DC) and Duff Moisture Code (DMC), accumulate the effects of precipitation and drying over multiple days. Consequently, even without explicitly including lagged precipitation as an input, the target variable (FWI) inherently reflects the influence of prior weather conditions. This allows the model to learn these dependencies implicitly from the current meteorological inputs.*

---

## Author Comment (AC2)

*We sincerely thank the reviewer for taking the time to review our manuscript. We believe their feedback has improved the clarity of the manuscript and overall quality of this work.*

In this study, the authors demonstrate the ability of deep learning (DL) models to emulate the Fire Weather Index (FWI) at 12 UTC. Specifically, three DL models are trained using either daily means or proxy data (as in Bedia et al., 2014) of weather variables relevant to FWI computation, to produce noon-time FWI estimates based on ERA5-Land. The authors also apply interpretability techniques to rank input variables according to their relevance in producing the FWI output. They find that, in high and extreme FWI scenarios, 24-hour accumulated precipitation is not needed to obtain accurate FWI values.

I appreciate the motivation behind this work and the authors' methodology. The results are compelling and well presented. However, I believe a deeper analysis in certain areas would significantly enhance the value of the paper.

**Major Comment 1: Generalization to datasets other than ERA5-Land**

One of the main motivations of this work is to emulate reference FWI conditions (i.e., those computed using 12 UTC weather variables and 24-hour accumulated precipitation) using proxy data from datasets that typically provide only daily information, such as climate model outputs. However, the authors do not show an example of applying their DL models to such external datasets.

Given that DL models are often sensitive to the data distribution used during training, applying the trained models to daily means from a dataset different from ERA5-Land (e.g., GCMs or other reanalysis) may yield inaccurate FWI emulations. Potential issues include discrepancies in statistical properties (e.g., mean, variance, extremes), spatial resolution (important for CNNs), or temporal characteristics.

I suggest the authors assess how their models perform when applied to an alternative dataset to emphasize the potential of this approach for correcting FWI estimates in climate simulations lacking noon-time fields.

*We appreciate the referee's thoughtful comment and fully agree that assessing the performance of the proposed models on other datasets, such as GCM outputs, is an important future step. In this study, however, we focus solely on emulation: that is, learning the transfer function with a DL model using the same database for both predictor set and predictand (ERA5-Land). Our primary goals are to evaluate the ability of DL models to emulate a multivariate index (FWI, in this case) from linked variables, setting a validation framework and looking for a reliable set up for the DL architectures and illustrating an intercomparison between the architectures, rather than to assess transferability across datasets (e.g., to other reanalyses or GCMs), which lies beyond the scope of this paper.*

*Nevertheless, we recognize the importance of transferability and are actively preparing a follow-up manuscript in which the proposed models will be applied to GCM outputs. In that*

*work, we will conduct a more detailed analysis of model performance on datasets with different statistical and temporal characteristics than ERA5-Land, and discuss the implications for correcting FWI estimates in climate simulations that lack noon-time meteorological fields. As noted earlier, we are also working on extending this line of research to GCM-based downscaling studies.*

*It is important to note that applying these models to other datasets, such as GCMs, requires careful consideration of statistical differences (e.g., biases in mean, variance, extremes, or spatial resolution), which would necessitate additional preprocessing (such as bias correction) and extensive validation. A rigorous evaluation of transferability therefore requires a more comprehensive and systematic study than can reasonably be accommodated within the scope of this manuscript.*

*To summarize, our study should be regarded as a case study on the ability of DL models to emulate FWI. While we see strong potential for their application beyond ERA5-Land, we consider it more appropriate to address transferability to other datasets (e.g., GCMs or alternative reanalyses) in future work. Given the already substantial length of the manuscript and the aforementioned scope, incorporating such an analysis here would not be feasible.*

**Major Comment 2: Temporal evaluation of FWI estimations**

**The authors conclude that their DL models capture both spatial and temporal variability of the reference FWI better than traditional proxy methods, and improve the detection of high-risk events. While the spatial evaluation is clearly presented, the paper does not seem to explicitly evaluate the temporal aspects of the DL-predicted FWI.**

**Beyond the Max Spell analysis, I suggest comparing the seasonal cycle of the reference FWI, proxy FWI, and DL-predicted FWI. This would help assess whether the models maintain consistent accuracy across different parts of the year or under seasonal biases. I also recommend extending the test dataset beyond the current 3-year window—e.g., from 2018 to 2024—to ensure more robust temporal assessment.**

*We thank the reviewer for the suggestion regarding the evaluation of temporal aspects of the FWI predictions. In addition to the spatial assessment, we provide an analysis of the monthly boxplots of the reference FWI, proxy FWI, and the DL-predicted FWI (Dense, DeepESD, U-Net) per climatological region, as shown in Figure 1. Each boxplot represents the distribution of FWI values for a given month over the 2012–2021 period, capturing both the median and the spread of the values.*

*This visualization demonstrates that the DL models not only reproduce the spatial patterns of FWI but also effectively capture the temporal variability across the year and the different regions. The median and interquartile ranges of the DL predictions closely follow the reference FWI throughout the seasonal cycle, including the high fire danger summer months,*

*whereas the traditional proxy FWI tends to slightly underestimate extremes during peak months (June–September). This analysis confirms that the DL models maintain consistent accuracy across different parts of the year and do not introduce significant seasonal biases.*

[Figure]

*Figure 1: Monthly boxplots for the reference FWI, proxy FWI, and the DL-predicted FWI (Dense, DeepESD, UNet) for CONTM (top-left), COASM (top-right) and ATL (bottom) regions (see Figure A1 of the manuscript to understand the climatological regions). Each boxplot represents the distribution of FWI values for a given month over the test period, capturing both the median and the spread of the values.*

*The discussion provided in this comment about the temporal evaluation and the Figure 1 have been added to the manuscript in Section 3.2.1.*

*Moreover, as suggested by the referee, we have extended the test period from 2018–2021 to 2012–2021 in order to provide a more comprehensive evaluation of the method's temporal robustness. By increasing the length of the test period, we are able to assess model performance across a wider range of interannual variability and climatic conditions, which allows for a more rigorous validation. The results obtained for the extended period are consistent with those observed in the original 2018–2021 test window, indicating that the method maintains its accuracy and reliability over a longer timeframe. This outcome reinforces the robustness of our approach and provides additional confidence in the generalizability of the model for FWI emulation.*

[Figure]

*Figure 2: Results from the Dense, DeepESD and U-Net model trained with the P0 predictors set (see Table 1 from manuscript) . The maps display differences relative to the reference FWI for the fire season (June–September) during the test period (2012–2021) for the FWI MAE (first row), FWI95 MAE (second row), Frequency FWI95 Relative Bias (third row) and Maximum annual Spell FWI 95 bias. The MAE value inside the map represents the spatially aggregated mean absolute error of the deep learning predictions with respect to the FWI reference, while Bias denotes the spatially averaged bias in absolute value.*

[Figure]

*Figure 3: Results from the U-Net model trained with the P0, P1 and  predictors sets (see Table 1 from manuscript) . The maps display differences relative to the reference FWI for the fire season (June–September) during the test period (2012–2021) for the FWI MAE (first row), FWI95 MAE (second row), Frequency FWI95 Relative Bias (third row) and Maximum annual Spell FWI 95 bias. The MAE value inside the map represents the spatially aggregated mean absolute error of the deep learning predictions with respect to the FWI reference, while Bias denotes the spatially averaged bias in absolute value.*

*Therefore, as previously commented, we obtain robust results independently of modifications in train (from 1979-2017 to 1979-2011) or test (from 2018-2021 to 2012-2021) period. The values of spatially aggregated MAE and bias of Figure 2 and 3 are consistent with the ones in the manuscript (Figures 3 and 6 respectively). Also the spatial patterns observed in the validation indices maps are similar, with a few exceptions such as the case of the Relative Bias of the FWI95 Frequency for the U-Net model trained with the P0 and P1 pattern. Accordingly, we mention in the new version of the manuscript in Section 2.5 that "The final results are presented for an independent test period spanning 2018–2021. We also evaluated longer test periods (2012–2021), which required shortening the training phase to 1979–2011. Since these tests produced robust and consistent results comparable to those for 2018–2021, they are not included in the text".*

**Minor 1:**

**The authors justify using the term "reference FWI" instead of "ground truth" because ERA5-Land is not observational. Since the DL models are trained on ERA5-Land, they likely inherit its biases. I suggest briefly discussing known limitations of ERA5-Land compared to observations, especially if no comparison with observed FWI is**

**included. This is specially important considering major comment 1 in which the ERA5Land's biases learned by the model might be propagate to other models.**

*Thank you for the comment. We will include in the discussion the limitations of ERA5-Land compared to the observations adding as appendix the biases between observational FWI data from the Spanish Agency of Meteorology (AEMET) and the FWI resulting from ERA5-Land computations. Here, we present the assessment about the limitations of ERA5-Land FWI compared with the observational data in terms of FWI and FWI95 climatologies biases.*

*In the new Appendix B: ERA5-Land limitations we have added the following text and Figure:*

*"In this section, we highlight the existent limitations in using ERA5-Land data in our analysis due to the inherited biases in ERA5-Land compared with observation data. These biases reflect systematic deviations from ground-based observations and can affect the reliability of the dataset for certain applications. Therefore, although ERA5-Land provides a valuable, spatially and temporally consistent climate dataset, its outputs should be used with caution and validated against local observations whenever possible.*

*In Figure B1, we illustrate the ERA5-Land biases in some stations in Spain with respect to observation data provided by the Spanish Agency of Meteorology (AEMET)."*

[Figure]

[Figure]

*Figure 4: Biases between observational FWI data from the Spanish Agency of Meteorology (AEMET) and the FWI resulting from ERA5-Land computations. The map in the left indicates bias for the mean FWI, while the map in the right indicates it for the FWI95.*

**Minor 2:**

**Please provide the actual thresholds used by the Spanish Meteorological Agency (AEMET) for fire danger classification, as these can vary by country and are important for interpretation.**

*Thank you for your comment. We will include the AEMET fire danger threshold in the manuscript, as it is necessary for understanding the FWI magnitudes associated with each category. We are also considering adding a table to the manuscript that briefly summarizes the thresholds for each category. We have added this table in the manuscript in Section 2.5.*

*Table: FWI Classes According to AEMET (Based on Percentiles)*

| Level | FWI Percentile Range |
|---|---|
| Low | Below 40th percentile |
| Moderate | 40th – 65th percentile |
| High | 65th – 85th percentile |
| Very High | 85th – 95th percentile |
| Extreme | Above 95th percentile |

**Minor 3:**

**I suggest moving Figure 1 to the Supplementary Information, as similar architectures have already been described in previous literature.**

*We thank the referee for the suggestion. Although it is true that similar architectures have been described in the previous literature, we strongly believe that including this figure makes the study more self-contained and improves reader comprehension. For readers who are not familiar with Deep Learning, it may be difficult to fully grasp the architecture from the textual description alone. Moreover, having the figure available directly in the manuscript makes the presentation clearer and more accessible.*

**Minor 4:**

**Consider merging Figures 2 and 3 to highlight the comparison between reference FWI, proxy FWI, and the DL emulators in a more compact and interpretable format.**

*We appreciate the referee's observation. However, we believe that merging Figures 2 and 3 would result in a very dense and large figure, reducing clarity. Additionally, we think the figures should remain separate because, first, we present our reference—the proxy commonly used in the literature—and then the bias between these two approaches. Furthermore, throughout the manuscript, validation figures such as Figures 3 and 6 are used to illustrate model bias or errors with respect to the true target values. In our view, this separation makes it easier to follow the narrative, identify issues, and clearly understand the results.*

**Minor 5:**

**It's unclear why the Freq. FWI95 for 12 UTC is not shown in Figure 1, even though it is later used to compute biases. Including it would help clarify the comparison.**

*Thank you for your comment. Although the frequency of FWI95 events is a key metric in validation, it is not explicitly shown in Figure 2 because, by construction, each grid point in the reference data records exactly 5% of days above its local 95th percentile threshold during the season. In other words, the value in each grid across the spatial map is 0.05. We have clarified this in the new version of the manuscript:*

*Section 3.1: "The FWI95 frequency for the reference FWI is not shown, because, by construction, each grid point in the reference data records exactly 5% of days above its local 95th percentile threshold during the season. Therefore, the value in each grid across the spatial map is 0.05."*

**Minor 6:**

**Could the overestimation of Freq. FWI95 by the UNet be explained by a general overestimation of FWI in this model, as suggested by the scatter plot? Are all three DL models trained on exactly the same days and years?**

*Yes, all three DL models (Dense, DeepESD, and UNet) were trained on exactly the same days and years and also with the same parameters setup, ensuring a fair comparison. Regarding the overestimation of the frequency of FWI95 by the UNet, this effect is due to an error in the calculation of the Figure. The correct version is the following attached below, therefore now showing a similar spatial pattern than the other DL models:*

[Figure]

*Figure 5: Results from the U-Net model trained with the P0 for the FWI95 Frequency Relative Bias. The maps display differences relative to the reference FWI for the fire season (June–September) during the test period (2018–2021) for the FWI MAE (first row). The Bias shown inside the panel denotes the spatially averaged bias in absolute value.*

*We thank the referee for the observation and this problem will be solved modifying the affected figures in the main manuscript. Despite this, the manuscript's storyline remains unchanged*

**Minor 7:**

**Please use either "proxy FWI" or "Proxy FWI" consistently throughout the text.**

*Thank you for your observation. We have replaced "proxy FWI" by "Proxy FWI" to be consistent throughout the text.*

**Minor 8:**

**Have you normalized the input data before training the DL models? If so, please specify the normalization method used.**

*Yes, we normalize our input data following a standardization. We set up the standardization as follows:*

$$x_i' = (x_i - \mu_i)/\sigma_i \quad \text{where i is the corresponding gridpoint}$$

*, where $x'$ represents the standardized value, $x$ represents the raw (unstandardized) value, $\mu_i$ and $\sigma_i$ represents the mean and standard deviation at gridpoint i. Parameters mu and sigma have been computed relative to the training period 1979-2011 (per gridpoint).*

*When the model is applied to unseen (test) data, the standardization is performed using the same $\mu_i$ and $\sigma_i$ values derived from the training data. This ensures that the test data is normalized in a way that is consistent with the training data, preventing information leakage and maintaining comparability between training and prediction phases.*

*A brief explanation has been added to the manuscript in Section 2.4.*

**Minor 9:**

**Since your DL architectures do not incorporate temporal dependencies, they may miss the effect of temporal accumulation in the Duff Moisture and Drought codes (e.g., DC, DMC). Why did you choose non-recurrent architectures over those incorporating temporal structure (e.g., LSTMs)?**

*We thank the referee for the constructive comments. In response, we implemented a ConvLSTM model with a temporal window of seven days (i.e., to predict day i, the model incorporates information from days i–6 through i). Given the substantial computational demands of ConvLSTM training, this implementation was carried out on GPUs. The ConvLSTM model adopts the general U-Net framework described in the manuscript, but substitutes the 2D convolutional layers with 2D ConvLSTM layers. Replicating the full depth of the original U-Net architecture was not feasible due to memory constraints; therefore, the depth of the ConvLSTM model was reduced accordingly.*

*Figure 6 presents a comparison between the ConvLSTM results and those of the model described in the manuscript for the JJAS season during the test period (2012–2022). The training parameters and conditions were kept identical to ensure comparability. Across the principal validation metrics reported in the paper, the ConvLSTM exhibits inferior performance relative to the other models. This outcome does not necessarily imply that ConvLSTMs are unsuitable for emulating the FWI; rather, it suggests that achieving competitive performance would require further optimization and the design of a more complex ConvLSTM architecture.*

*Such an investigation is currently underway, as part of ongoing work aimed at optimizing time-dependent models for FWI downscaling applications. However, these efforts fall beyond the scope of the present study, which is focused on FWI emulation. Moreover, we consider that the use of ConvLSTM is not strictly necessary to capture temporal dependencies, as these are inherently embedded in the computation of the FWI itself. As the referee rightly*

*noted, several components of the FWI—such as the Drought Code (DC) and the Duff Moisture Code (DMC)—explicitly account for the temporal evolution of moisture and drought conditions. Therefore, the FWI, as the target variable, already encapsulates this temporal characterization. Consequently, we consider that the absence of explicit temporal modeling in the network architecture is not critical, as the models are learning a function that implicitly includes these dependencies.*

[Figure]

*Figure 6: Results from the Dense, DeepESD, U-Net and Conv-LSTM model trained with the P0 predictor set (see Table 1 from manuscript) . The maps display differences relative to the reference FWI for the fire season (June–September) during the test period (2012–2021) for the FWI MAE (first row), FWI95 MAE (second row), Frequency FWI95 Relative Bias (third row) and Maximum annual Spell FWI 95 bias. The MAE value inside the map represents the spatially aggregated mean absolute error of the deep learning predictions with respect to the FWI reference, while Bias denotes the spatially averaged bias in absolute value.*

*In the manuscript we have clarified that we have tested a ConvLSTM, but it is not included in the text in Section 2.4: "Alternative architectures, such as Convolutional Long Short-Term Memory (ConvLSTM), were also evaluated. However, owing to their inferior performance relative to the selected architectures and their greater computational demands, they are not presented in this manuscript."*

**Minor 10: Interpretation of input variable relevance**

**The saliency maps suggest that precipitation is only relevant in low-FWI scenarios. However, this may be a result of how precipitation contributes to the FWI calculation**

itself—namely, it offsets the fuel dryness components. Therefore, in high and extreme FWI events (which typically occur during dry periods), the precipitation input often has a value of zero, contributing little additional information for the DL model.

It would be insightful to give more information about why the model changes its focus depending on the region and the type of FWI (non or extreme value). Otherwise, the only information that this result gives us is that for the DL temperature, relative humidity and wind speed are sufficient for obtaining accurate high and extreme FWI events. But, is this true in reality?

Considering this work uses ERA5Land, the predictor variables (T, RH, P and ws) are non independent from each other. In fact, temperature and dew point temperature (needed to compute relative humidity) are variables calculated by the land surface model in ERA5Land, while total precipitation and wind components are forcing variables interpolated from ERA5. Therefore it is likely probable that temperature and relative humidity in ERA5Land reflects the effects of changes in total precipitation and wind speed and, therefore, these two last variables are not so needed by DL models. This is just a guess and it likely won't be the full explanation of your interpretability results... but in any case it would be very valuable to give more information about this or at least mention it if you agree on this limitation.

*We thank the referee for this very thoughtful and detailed comment. We agree that the apparent low relevance of precipitation in high- and extreme-FWI scenarios is strongly linked to how precipitation contributes to the FWI itself. Specifically, precipitation primarily influences the fuel moisture codes, and when FWI values are high, these codes already reflect prolonged dry conditions, making the precipitation input frequently zero and thus uninformative for the DL model. We clarify this point in the revised manuscript to ensure that this mechanism is explicitly discussed.*

*Regarding the reviewer's second point, we also agree that providing more insight into why the model's feature attribution changes across regions and FWI regimes would strengthen the interpretability section. We will expand our discussion by emphasizing two aspects:*

*FWI definition dependency – The DL model's focus on temperature, RH, and wind in high/extreme cases mirrors the FWI's own reliance on these variables under dry conditions. However, this does not imply that precipitation (or other inputs) is unimportant in reality for fire risk; rather, it reflects the structure and sensitivity of the FWI metric that the DL model is trained to emulate.*

*Predictor interdependencies in ERA5Land – As the reviewer correctly notes, ERA5Land variables are not independent: RH is derived from temperature and dew point (affected by precipitation indirectly), while precipitation and wind are assimilated forcings. These dependencies likely explain why the DL model can achieve high predictive accuracy even when precipitation and wind receive lower attribution scores. We agree that this introduces a limitation in interpreting the saliency maps, since the apparent dominance of temperature and RH may partly arise from their embedded relationships with other drivers.*

*We will incorporate this limitation into the revised manuscript and make it clear that the interpretability results should not be read as definitive statements about the physical importance of individual meteorological drivers in real-world fire danger processes. Rather, they highlight how the DL model leverages the structure of the ERA5Land inputs and the FWI formulation itself.*

*Therefore, in* **Section 3.5** *(in the third paragraph of the discussion) we have incorporated the following explanation which summarizes the previous ideas:*

*"This finding is consistent with the definition of the FWI itself, which relies primarily on temperature, relative humidity, and wind speed under dry conditions. The DL model thus reflects the structure and sensitivity of the FWI metric it is trained to emulate. Importantly, this does not mean that precipitation (or other inputs) is irrelevant for real-world fire danger; rather, it highlights that the predictand (FWI) gives limited weight to precipitation in high and extreme danger situations.*

*Moreover, ERA5-Land predictor variables are not independent. Relative humidity, for instance, is derived from temperature and dew point, which are indirectly influenced by precipitation, while precipitation and wind are assimilated forces. These interdependencies likely contribute to the model's ability to achieve high predictive accuracy even when precipitation and wind receive lower attribution scores."*

*For all these reasons, we thank the referee for this comment, which enriches the discussion of the interpretability results in the manuscript.*

**Minor 11:**

**I miss an experiment in which you assess how the DL models learn to compute the reference FWI using input variables at 12 UTC. Your experiments P1 and P2 address this question to some extent, but the resulting biases could also arise from difficulties relating daily aggregates to FWI 12 UTC data. It may be worth including this experiment to provide insight into where the obtained biases in the DL models may come from.**

*We thank the referee for this insightful suggestion. In response, we have included an additional sensitivity experiment designed to evaluate the ability of the deep learning models (deepESD, fully connected (dense) networks, and U-Net architecture) to learn the transfer function that defines the Fire Weather Index (FWI) using input variables at 12:00 UTC, consistent with the temporal resolution of the reference index. This new experiment is now included in the revised manuscript in Figure D1 as part of a new Appendix D: Sensitivity analysis of deep learning models and evaluation of other predictors sets.*

*This experiment complements our previous configurations (P1 and P2) by isolating the effect of temporal aggregation. Specifically, we compare model performance when trained with instantaneous inputs (i.e., values at 12:00 UTC and 24-hour precipitation) versus daily mean inputs. This comparative framework allows us to disentangle the intrinsic biases of each modeling approach from the additional error introduced by using daily-aggregated predictors.*

*Our results, summarized in Figure 7 below, show that models trained with instantaneous inputs exhibit lower bias and improved accuracy, while maintaining a similar spatial error pattern. This confirms that part of the error observed in experiments P1 and P2 stems from the mismatch in temporal resolution between the predictors and the reference FWI.*

*Furthermore, we find that the U-Net architecture yields the lowest intrinsic bias in emulating the actual FWI function. It consistently exhibits the smallest biases in both FWI and FWI95, as well as in the predicted frequency of FWI95 events and the mean annual maximum duration of FWI95 spells. These results suggest that the U-Net architecture may offer enhanced generalization capabilities. Its ability to maintain low bias across both instantaneous and aggregated inputs indicates robustness to temporal variability, which is a key factor in modeling climate indices.*

[Figure]

*Figure 7: Results from the Dense, DeepESD and U-Net model trained with 12UTC input variables (temperature, 24h-accumulated precipitation, relative humidity and wind speed) . The maps display differences relative to the reference FWI for the fire season (June–September) during the test period for the validation indices. The MAE value inside the map represents the spatially aggregated mean absolute error of the deep learning predictions with respect to the FWI reference, while Bias denotes the spatially averaged bias in absolute value.*

*A discussion about these results is provided in the new version of the manuscript in Section 3.2.1: "Before discussing the performance of the DL models across the validation indices, we first highlight how daily aggregation of the input data affects model performance. Figure D1 in Appendix D presents the results from the Dense, DeepESD, and U-Net models trained using 12:00 UTC input variables (temperature, 24-hour accumulated precipitation, relative humidity, and wind speed) to compute the FWI. This sensitivity experiment evaluates the models' ability to learn the transfer function defining the FWI using inputs at 12:00 UTC, consistent with the temporal resolution of the reference index.*

*This analysis complements our other model configurations by isolating the effect of temporal aggregation. Specifically, we compare performance when models are trained with instantaneous inputs (i.e., values at 12:00 UTC and 24-hour precipitation) versus daily mean inputs. This framework allows us to separate the intrinsic biases of each model from the additional error introduced by using daily-aggregated predictors. Figure D1 shows that models trained with instantaneous inputs exhibit lower bias and improved accuracy while maintaining a similar spatial error pattern. This confirms that part of the error observed in experiment P0 (Figure 3) stems from the mismatch in temporal resolution between the predictors and the reference FWI.*

*However, some regions, such as the Mediterranean areas for FWI MAE and the Cantabrian Mountains, the Pyrenees, and the Mediterranean coast for FWI MAE95, exhibit intrinsic errors even when instantaneous inputs are provided to the DL models.*

*Moreover, the U-Net architecture demonstrates the lowest intrinsic bias in emulating the actual FWI function. It consistently shows the smallest biases in FWI and FWI95, as well as in the predicted frequency of FWI95 events and the mean annual maximum duration of FWI95 spells. These results suggest that U-Net offers enhanced generalization capabilities. Its ability to maintain low bias across both instantaneous and aggregated inputs indicates robustness to temporal variability, a key factor in modeling climate indices."*